# Prediction of Aggregation of Biologically-Active Peptides with the UNRES Coarse-Grained Model

**DOI:** 10.3390/biom12081140

**Published:** 2022-08-18

**Authors:** Iga Biskupek, Cezary Czaplewski, Justyna Sawicka, Emilia Iłowska, Maria Dzierżyńska, Sylwia Rodziewicz-Motowidło, Adam Liwo

**Affiliations:** Faculty of Chemistry, University of Gdańsk, Wita Stwosza 63, 80-308 Gdańsk, Poland

**Keywords:** peptide aggregation, molecular dynamics, replica exchange, UNRES model

## Abstract

The UNited RESidue (UNRES) model of polypeptide chains was applied to study the association of 20 peptides with sizes ranging from 6 to 32 amino-acid residues. Twelve of those were potentially aggregating hexa- or heptapeptides excised from larger proteins, while the remaining eight contained potentially aggregating sequences, functionalized by attaching larger ends rich in charged residues. For 13 peptides, the experimental data of aggregation were used. The remaining seven were synthesized, and their properties were measured in this work. Multiplexed replica-exchange simulations of eight-chain systems were conducted at 12 temperatures from 260 to 370 K at concentrations from 0.421 to 5.78 mM, corresponding to the experimental conditions. The temperature profiles of the fractions of monomers and octamers showed a clear transition corresponding to aggregate dissociation. Low simulated transition temperatures were obtained for the peptides, which did not precipitate after incubation, as well as for the H-GNNQQNY-NH_2_ prion–protein fragment, which forms small fibrils. A substantial amount of inter-strand *β*-sheets was found in most of the systems. The results suggest that UNRES simulations can be used to assess peptide aggregation except for glutamine- and asparagine-rich peptides, for which a revision of the UNRES sidechain–sidechain interaction potentials appears necessary.

## 1. Introduction

Peptides and peptidomimetics have since long been used as drugs, which act as enzyme inhibitors, agonists or antagonists of hormones or neurotransmitters or induce immunoresponse [1,2,3]. It is estimated that the Peptide Therapeutics Market is going to reach $50.60 billion by 2026 [4]. Peptides as therapeutics have many advantages, including high selectivity for target molecules and good biocompatibility with the human organism. Moreover, owing to their smaller size, they exhibit better tumor penetration in comparison to monoclonal antibodies [5].

Good solubility in water and disinclination to aggregation are often required for a designed peptide to become a drug candidate [6,7,8]. Peptide solubility and tendency to aggregation in water can be assessed quickly based on the fraction of polar and charged residues and the estimated isoelectric point [9]. However, this method provides only a cursory estimation and the actual peptide–peptide interactions should be considered to determine whether the aggregates are stable under any given conditions.

Peptide aggregation can result in the formation of amyloid deposits in the nerve tissue, causing a variety of neurodegenerative diseases, such as Parkinson’s and Huntington’s diseases and familial amyloidoses [10].

In living organisms, the peptides formed by the enzymatic digestion of proteins (e.g., the amyloid-beta, A$\beta_{42}$ peptide cleaved from the amyloid-precursor protein [11], polyglutamine chains cleaved from Huntington [10], the repeat domains of the Tau protein (TauRD) [12], etc.) typically form amyloid deposits. Even though a number of measures have been attempted to prevent amyloidoses, including the use of nanoparticles [13], the problem is far from being solved. On the other hand, the aggregation of designed peptides can be beneficial, enabling us to design new materials [14] and gels [15].

Molecular-simulation methods have long been used to study peptide aggregation, including both all-atom [16,17,18] and coarse-grained (CG) methodologies [19]. Of those, the methods based on CG models offer much longer time- and size-scales, including the possibility of simulating aggregation from scratch, even though this extension is achieved at the inevitable expense of modeling accuracy. Many dedicated CG models, e.g., the Bereau–Deserno model [20], have been developed to study aggregation, including those dedicated to study the glutamine-rich systems [21]. The CG models also include minimal ones, designed to study the general features of the aggregation process [22].

In this study, we used the physics-based UNRES coarse-grained model [23,24,25,26] developed in our laboratory to investigate the association of 16 peptides, whose aggregation has already been studied experimentally [27,28,29,30,31,32,33,34], as well as of four new peptides. These peptides have sizes ranging from 6 to 32 amino-acid residues and varying contents of hydrophobic, polar and charged residues. Of those, 12 are hexa- or heptapeptides excised from the β-sheet and loop regions of human cystatin C [27,34], the Tau protein [28,33], or the Sup36 prion protein [29,30] and are potentially aggregation-prone.

The other ones contain potentially aggregating sequences merged with charged sequences with the intent of making functionalized gels or films or are signaling peptides [31,32]. Even though the aggressive coarse-graining has been implemented in UNRES (only two sites per residue), it is capable of ab initio modeling the structure, dynamics, thermodynamics and free-energy landscapes of small and medium-size proteins and bioninformatics- and data-assisted modeling of large proteins, including oligomeric proteins [23,26].

It is regularly being tested in the biannual Community Wide Experiments on the Critical Assessment of Techniques for Protein Structure Prediction (CASP) [35]; see Ref. [36] for recent results. It has also been used with success to study the aggregation of the Aβ amyloidogenic peptides [37,38,39] and the association of the Aβ and Tau peptides [40,41].

## 2. Materials and Methods

### 2.1. Peptides Studied

The sequences of the peptides studied, together with the total charges at pH = 7 estimated by the peptcalc.com utility [42], grand hydropathy indices (GRAVY), estimated based on the Kyte and Doolitle hydropathic indices [43] using the ExPasy [44] protparm tool (https://web.expasy.org/protparam/ accessed on 8 August 2022), experimental concentrations, box-side lengths used in simulations, aggregation information, the types of aggregates if formed and literature sources, if applicable, are collected in Table 1. As shown, of the 20 peptides considered, only three did not precipitate after the up to 21-day incubation period. The results of our simulations correspond to equilibrium ensembles, i.e., to an infinitely long incubation period.

### 2.2. Experimental Procedures

As shown in Table 1, the data of 13 of the peptides studies were taken from our previous work [27,31,32,34]. Three peptides (Tau R2/wt, Tau R3/-P312 and Sup35(7-13) were studied by other researchers [28,29,33]; however, we synthesized them and performed measurements in this work. The remaining four peptides (CysZ14, QIVFFA, KGHK-KGHK and IM-IM) have not been studied previously.

The seven peptides mentioned above—CysZ14, QIVFFA, Tau R2/wt, Tau R3/-P312, Sup35(7-13), KGHK-KGHK and IM-IM—were synthesized by means of the standard solid phase method using a Liberty BlueTM automated microwave synthesizer (CEM Corporation, Matthews, NC, USA). The syntheses were performed according to the Fmoc chemistry methodology using N−Fmoc protected amino acids.

The raw products were purified using semi-preparative column Jupiter Proteo C12 (Phenomenex, Torrance, CA, USA). The pure products were analyzed using analytical RP-HPLC and mass spectrometry (BIFLEX III MALDI-TOF, Bruker, Germany and LC-MS ESI-IT-TOF, Shimadzu, Japan). The detailed procedure of the synthesis steps, the concentrations of the coupling and deprotection reagents we used, as well as the purification conditions were described in our earlier work [31].

The peptides were dissolved by shaking the samples in water (applying ultrasound if necessary) at the temperature of 37 °C (310 K). Under these conditions, all peptides were soluble at the concentrations specified in Table 1.

Circular dichroism (CD) measurements were made for the peptide solutions before and after incubation, except for RADA-IM—RADA-KGHK, which formed gels almost immediately, and for the non-associating KGHK-KGHK and IM-IM peptides. After the incubation period, the peptide precipitate was removed by centrifugation prior to recording the CD spectra of the peptides remaining in solution.

The CONTIN program from the CDPro package [46] was used to estimate the fractions of extended, α, turn and statistical coil peptides. The calculated fractions are summarized in Table 2. Transmission Electron Microscopy (TEM) was used to determine the shapes of the aggregates.

### 2.3. Coarse-Grained Simulations

The coarse-grained simulations were conducted with the UNRES model of polypeptide chains [23,26], using the scale-consistent NEWCT-9P variant of the UNRES force field developed and parameterized in our recent work [24,25]. Briefly, UNRES is a highly coarse-grained model, in which a polypeptide chain is represented as a sequence of α-carbon (C${}^{\alpha }$) atoms, with united sidechains attached by virtual bonds to the respective C${}^{\alpha }$ atoms and the united peptide groups located in the middle between two consecutive C${}^{\alpha }$ atoms. Only the united peptide groups and united sidechains are interaction sites, with the C${}^{\alpha }$s serving to define the backbone geometry.

The solvent is implicit in the interaction potentials. The sites and the corresponding interaction potentials have axial and not spherical symmetry and multibody terms are present to couple the backbone-local and backbone-electrostatic interactions, which is necessary for the correct modeling of regular secondary structures. The present parameterization of the sidechain–sidechain interaction potentials corresponds to physiological pH. The different flexibility of glycine and of the proline residues is accounted for in the backbone-local and backbone-multibody potentials [25]. UNRES can also handle dynamic formation and the dissociation of disulfide bonds [47].

The details of the model and the recent variant of the force field can be found in the references cited [23,24,25,26]. We used the parallelized UNRES program [48], which was recently heavily upgraded and optimized [49]. For each peptide, the simulations were conducted for eight chains in a cubic periodic box at a constant volume. Consequently, the peptide concentration was fixed at the respective experimental values for each system studied. The side of a box corresponded to the experimental peptide concentration. The box-side lengths are summarized in Table 1.

We chose to simulate only eight-chain systems because achieving convergence of the ensembles was critical in the determination of the aggregate-dissociation temperatures and, therefore, in the assessment of whether a peptide aggregates at the experimental sample-incubation temperature. This setup is similar to that of the protocol recommended for all-atom simulations with GROMACS [50], in which six molecules are placed in a cubic periodic box with the 100 Å side [17].

We conducted multiplexed replica exchange molecular dynamics (MREMD) [51] simulations, using Lagrange molecular dynamics and the MREMD extension implemented in UNRES [52,53,54,55], which were recently optimized in the revised UNRES package [49]. It should be noted that MREMD is a method for searching the conformational space and, consequently, to determine equilibrium conformational ensembles and ensemble averages, which is sufficient for the purpose of the present study. On the other hand, because each replica walks in the temperature space during the course of a simulation, MREMD is not appropriate to study the time evolution of a system.

For each systems, the replicas were run at 12 temperatures equal to 260, 272, 279, 284, 288, 291, 294, 298, 308, 322, 341 and 370 K, which were determined using the Hansmann algorithm [56] to provide the maximum number of walks in the temperature space. Each replica was multiplexed by 4, this giving a total of 48 trajectories for each system.

All simulations were run in the Langevin-dynamics mode, which provides a canonical ensemble, the time step being 9.78 fs. The viscosity of water was scaled by the factor of 0.02 to speed up the simulations. Replicas were exchanged every 10,000 MD steps, and snapshots were saved at each exchange point. The number of MD steps of each trajectory ranged from 40,000,000 to to 120,000,000 (from about 0.4 to 1.2 μs) for the hexapeptide systems and from 80,000,000 to 320,000,000 (from about 0.8 μs to about 3.2 μs) for larger peptides. Each trajectory was started from chains randomly placed in the box, with each chain in a randomly-generated conformation, subject to the condition of non-overlap.

Each simulation was run until converged. The convergence was checked by monitoring the temperature profiles of $f_{8}\left(T\right)$ (the fraction of octamers) (Equation (Equation 1)) averaged over the consecutive 10,000,000-steps (1000-snapshot) windows, taking every eighth snapshot from the respective trajectory window (a total of 48,000/8 = 6000 snapshots per window, for all 48 trajectories). For each system, the standard deviations of $f_{8}\left(T\right)$ ($\sigma_{f_{8}}$) at the subsequent simulation windows (except for the last one) from the last window were calculated and plotted in window index. A simulation was considered to be converged if $\sigma_{f_{8}}$ stopped to decrease. A sample plot for the CysZ3 and the RADA-KGHK systems is shown in Figure 1.

It should be noted that, owing to the elimination of the fast degrees of freedom in UNRES, a time unit of UNRES simulations corresponds to about 1000 time units of all-atom simulations [49,53] and, consequently, the millisecond time scale was reached in our simulations. Therefore, for each trajectory, the time covered by our simulations is effectively 400–3200-times greater than that recommended in the GROMACS-based all-atom protocol [17]. The search of the conformational space is further accelerated by MREMD. Nevertheless, even with this extension, the simulation time-scale is far from the experimental incubation time, and, consequently, the convergence of the ensembles could only be achieved given the small number of chains.

### 2.4. Analysis of Simulation Results

After convergence was achieved, the last 1000 snapshots from each trajectory with the sampling frequency of 8 (a total of 6000 snapshots) were taken for further analysis, which was performed using our implementation [54] of the binless Weighted Histogram Analysis Method [58]. WHAM enabled us to compute the temperature profiles of the ensemble-averaged quantities and to determine the conformational ensembles at selected temperatures, which were the room temperature, at which the CD measurements were made and the incubation temperature, respectively.

The conformational ensembles were dissected into five clusters by means of Ward’s minimum-variance method [59]. The clusters were ranked by the cumulative probabilities of the constituent conformations, as described in our earlier work [54]. The structure closest to the mean structure of the respective cluster was selected as the representative of the entire cluster. Selected structures were converted to the all-atom structures using the PULCHRA [60] and SCWRL [61] algorithms.

To analyze the dependence of peptide association on temperature, we computed the temperature profiles of the fractions of isolated (monomeric) chains and chains in *m*-chain oligomers, $f_{m}\left(T\right)$, where m=1 for a monomer (Equation (Equation 1)).
(1)$$f_{m}\left(T\right)=\sum_{i=1}^{N}w_{i}\left(T\right)f_{mi}$$
where $f_{mi}$ is the fraction of chains in the *m*-chain oligomer in snapshot *i*, $w_{i}\left(T\right)$ is the weight of snapshot *i* calculated with WHAM [54,58], and *N* is the number of snapshots in the batch. For each temperature, the conformational weights are normalized to 1 (Equation (Equation 2)).
(2)$$\sum_{i=1}^{N}w_{i}\left(T\right)=1$$

The oligomers were identified, and chains were assigned to the subsequent oligomers using the following procedure. For a given snapshot, the search for oligomers was started from the first peptide chain. Chains from 2 to 8 were examined, and the first one that was found associated with chain 1 was added to form a dimer. Two peptide chains were considered associated if the distance between any C${}^{\alpha }$ atom of the first and any C${}^{\alpha }$ atom of the second chain was shorter than seven Å. If no chain was associated with the first one, the search was restarted taking the chain with index 2 and the subsequent ones.

Once two chains were found associated, the remaining chains were looped over to find the first one, which was associated with any of the two. The procedure continued until no more peptide chains were found associated with any chain in the oligomer. This procedure resulted in the identification of the first oligomer (which could be a monomer, if the first chain was not associated with the other chains). The procedure was applied iteratively to the peptide chains which were not included in the previously found oligomers, until all chains were assigned to the respective oligomers or found to be isolated.

To determine the character of the aggregates, we computed the temperature profiles of the content of statistical-coil ($f_{c}\left(T\right)$), β-sheet [$f_{\beta }\left(T\right)$] and α-helical [$f_{\alpha }\left(T\right)$] structure, as defined by Equation (Equation 3).
(3)$$f_{s}\left(T\right)=\sum_{i=1}^{N}w_{i}\left(T\right)f_{sij}$$
where *s* equals *c*, β or α and $f_{sij}$ is the fraction of residues in state *s* for chain *j* and snapshot *i*. We used the algorithm of our earlier work [62] to determine, from the coarse-grained geometry, if a residue is in the *c*, β or α state. This algorithm takes into account the backbone-peptide-group-interaction pattern (identified with hydrogen-bond network) and local main-chain geometry.

## 3. Results and Discussion

### 3.1. Peptide Aggregation

The temperature profiles of the fractions of monomers and associates from dimers to octamers, obtained by MREMD simulations of eight-chain systems are shown, for selected peptides (CysZ5, CysZ8 and RADA-IM), in Figure 2a,d,g. It can be seen from the figure that octamers and monomers dominate before and after the transition temperature, respectively. The dimers constitute a remarkable fraction of the ensemble after the transition temperature, particularly for the RADA-IM—RADA-KGHK peptides.

For each system, the $f_{8}\left(T\right)$ curve has a sigmoidal shape with an inflection point corresponding to the respective heat-capacity ($C_{v}$) peak (panels c, f and i of the figure)—being indicative of the first-order melting transition. It must be noted that, because the solvent is implicit in UNRES, the heights of the heat-capacity peaks are exaggerated. Consequently, the $C_{v}\left(T\right)$ curves can be analyzed only qualitatively.

The minor peaks at lower temperatures correspond to melting of the β-structure. The oligomer-melting transition is particularly sharp for RADA-IM (Figure 2g), as well as for RADA-GHK and RADA-KGHK. This result is in qualitative agreement with the experiment because all three RADA-based peptides form gels immediately after dissolving, as opposed to the other peptides for which aggregation can take up to 3 weeks (however, the fibrils start to appear already after the first 3 days of incubation).

The observation that the simulated aggregation of the peptides under study is effectively an all-or-none transition is supported by the fact that the fractions of the secondary structure of the peptide solution before and after incubation estimated from CD measurements do not change significantly for all peptides under study but CysZ4 (Table 2), this suggesting that monomers only are present in solution. For CysZ4, the amount of extended structure increases remarkably after incubation, which suggests that some oligomers are present in solution.

In Figure 3a, the mid-point temperatures of octamer dissociation ($T_{8m}$) are shown for all of the systems studied. The $T_{8m}$ values moderately correlate with the GRAVY indices collected in Table 1. The respective plot is shown in Figure 4. Such a correlation could be expected, because the propensity to association is largely determined by peptide hydrophobicity. On the other hand, the RADA-based and the FC-based peptides, which contain many charged residues and are prone to gel or fibril formation, respectively (Table 1), exhibit significantly higher $T_{8m}$ than could be expected based on their hydropathic indices. This observation suggests that our simulations could capture finer aspects of aggregation propensity, such as the cooperativity between side-chain-contact and interstrand backbone-hydrogen-bond formation and local chain structure.

In panel b of Figure 3, the fractions of oligomers and that of the monomer at *T* = 310 K (at which the peptide solutions were incubated) are shown in the form of stacked-histogram plots. It can be seen that the mid-point temperatures are the lowest for CysZ1, Sup35(7-13), KGHK-KGHK and IM-IM. Of those, CysZ1, KGHK-KGHK and IM-IM do not form any precipitates (cf. Table 1), while Sup35(7-13) forms small fibrils and was also reported to form microcrystals in the presence of Zn(II) or Cd(II) cations [29,30]. The $T_{8m}$ values are also several degrees below the incubation temperatures for CysZ2 and CysZ3, which form small fibrils after a long incubation period. The fractions of octamers at the incubation temperatures follow the values of $T_{8m}$, being zero for CysZ1, Sup35(7-13), IM-IM and KGHK-KGHK.

From Figure 3 it can be concluded that MREMD simulations of eight-peptide systems with UNRES give good predictions of the ability to association. The only exception is Sup35(7-13), for which UNRES predicts a low octamer mid-point dissociation temperature, while its sequence is a typical amyloid-forming sequence. The reason for this appears to be the presence of many glutamine and asparagine residues in the sequence of this peptide (cf. Table 1).

The UNRES sidechain–sidechain interaction potentials of these residues do not include multibody effects—namely, the ability of their amide groups to form strong hydrogen bonds with each other while dehydrated. This is likely to happen when many Gln or Asn residues are close in the amino-acid sequence (as, e.g., in the peptide fragments excised from huntingtin). Then, their side-chain amide groups can form close contacts and, consequently, become screened from the solvent. The present UNRES treats the side chains of Asn and Gln as polar non-associating side chains.

This observation strongly suggests that screening from the solvent and, likely, a contribution from solvent polarization, should be introduced in UNRES to handle the sidechain–sidechain interactions in systems with many glutamine and asparagine residues. An attempt at introducing solvent screening to improve the treatment of backbone-peptide-group interactions was made in our earlier work [63].

### 3.2. Secondary Structure of Peptide Aggregates

The temperature profiles of the secondary-structure content of selected eight-peptide systems (CysZ5, CysZ8 and RADA-IM) are shown in Figure 2b,d,f, respectively. The positions of the inflection points in the temperature profiles of the fractions of β structure and the amounts of the β-sheet, α-helical and statistical-coil structure at the room temperature (*T* = 298 K), at which the CD measurements were made, obtained from MREMD simulations, are shown in Figure 3a,c, respectively. The temperatures of the inflection points and not those of half-decay of the β-structure are reported because, as opposed to the fraction of octamers, the fraction of β-structure never reaches 1, even at the lowest simulation temperature.

From the temperature profiles (Figure 2b,e,h), it follows that the fraction of residues forming β-structure decreases with temperature. From the plots of $f_{\beta }\left(T\right)$ for the selected systems and from the $T_{i,\beta }$ values, it can be seen that, except for the RADA-IM—RADA-KGHK peptides and for those with low $T_{8m}$, the amount of β-structure decays at temperatures lower than the octamer-dissociation midpoint.

For some systems, e.g., CysZ5, the difference between the two temperatures is significant (Figure 2a,b), while for most of the systems, e.g., for CysZ8 (Figure 2d,e), the differences are smaller and the amount of β-structure at the room temperature is still significant. The different values of $T_{8m}$ and $T_{\beta ,i}$ are reflected in the presence of two heat-capacity peaks, one of which corresponds to the melting of β-structure and the other to octamer dissociation (Figure 2c,f).

The RADA-IM—RADA-KGHK systems are different from the other ones because the sharp decay of the amount of octamers nearly coincides with that of β-structure (Figure 2g,h and Figure 3). This feature is also reflected in the appearance of a single sharp heat-capacity peak (Figure 2i). It can be seen from Figure 3c that the α-helical structure is residual at the room temperature. For the RADA-IM—RADA-KGHK peptides, the α-structure emerges in quite a significant amount near $T_{8m}$, as displayed in Figure 2h for RADA-IM.

It should be noted that the amounts of extended and α+turn structures estimated from the CD spectra for dissolved peptides and for the RADA-IM— RADA-KGHK gels (Table 2) are higher than those of the β and α structures obtained from simulations. However, it should be noted that CD is sensitive to the local chain conformation and not to the actual presence of secondary-structure elements stabilized by hydrogen bonds.

Therefore, CD measurement suggest the presence of the extended structure even for the hexa- and heptapeptides studied in solution (where they are presumably monomers), which are unlikely to form intra-chain β-sheets. In our analysis of the simulated structures, a residue is assigned the β state only when the peptide group next to it exhibits strong mean-field interactions with another peptide group and belongs to a β-sheet pattern.

Similarly, the “helix” or “turn” conformation found by CD only indicates that a residue belongs to the α regions of the Ramachandran map, while in our analysis a residue is assigned the α structure only if the peptide group next to it belongs to the α-helix pattern. Nevertheless, in agreement with the results of our simulations, the fraction of extended structure from CD measurements exceeds that of α+turn structure for all of the systems studied.

### 3.3. Spatial Structure of the Aggregates

From the results discussed in Section 3.2 it can be concluded that the association into hydrogen-bonded β-sheets plays an important role in all systems that associate into octamers, except for CysZ5, for which the amount of β-structure is residual and, consequently, association occurs through hydrophobic interactions. Inspection of the representative strictures of the octamers of CysZ5 and CysZ8 at room temperature (Figure 5a,c) confirms that this is indeed the case, and that a regular amyloid-like structure is formed for CysZ8 but not for CysZ5.

Amyloid-like structures similar to that of CysZ8 were obtained at room temperature for all hexa- and heptapeptides except for CysZ1 and Sup35(7-13), which did not associate remarkably at this temperature. The scattered plots of the backbone ϕ and ψ angles shown in panels b and d of the figure indicate that the residues of CysZ8 reside in the extended region of the Ramachandran map, while those of CysZ5 are more scattered.

The ϕ and ψ angles were calculated from the all-atom structures converted to the all-atom representation from the coarse-grained structures (cf. Section 2.4). These structures do not, thus, result from explicit all-atom simulations. Nevertheless, for qualitative assessment of the local conformational states of the amino-acid residues, they seem to be sufficient. It should also be noted that all systems under study are highly flexible, especially with regard to chain-packing topology. Consequently, the structures presented in Figure 5, Figure 6 and Figure 7 represent only the most populated families and not the entire ensembles.

Except for RADA-IM—RADA-KGHK, the larger peptides (functionalized aggregating sequences) form irregular aggregates. As an example, the structure of the octamer of FC-GHK is shown in Figure 6a. As opposed to CysZ8 and CysZ5, the residues visit the whole accessible regions of the Ramachandran map, which confirms that the chain structure is flexible.

At the room temperature, the RADA-IM—RADA-KGHK peptides form a β-sheet structure, in which the (RADA)${}_{4}$ sections are involved in a surface β-sheet with the arginine and aspartic-acid residues stretching outside (Figure 7a). The whole structure is a distorted β-barrel. This structure can extend into a larger β-sheet accommodating a large amount of water because of the presence of charged surface residues, which explains the propensity of the RADA-IM—RADA-KGHK peptides to form gels [32].

At *T* = 310 K, the β-sheet becomes partially destroyed and part of the (RADA)${}_{4}$ sections form α-helices (Figure 7c). This observation suggests that the aggregates form micelles and the peptide solution turns into a sol as temperature increases. However, the experiment demonstrates that the gel is stable even at higher temperature, although the secondary structure becomes lost [32].

The observed changes of the tertiary structure resulting from heating the system from *T* = 298 K to *T* = 310 K are reflected in the plots of the ϕ and ψ angles shown in panels b and d of Figure 7. At *T* = 298 K, the extended region has the highest population, although the peptide visits all accessible regions of the Ramachandran maps while, at *T* = 310 K, the population of the helical region remarkably increases and that of the extended region decreases.

To determine the flexibility of peptide chains in octamer aggregates, we calculated the standard deviations of the C${}^{\alpha }$ atoms of the consecutive residues of all chains and all five models obtained by cluster analysis at *T* = 298 K from those of the mean chain structures (averaged over all chains and all five models obtained at *T* = 298 K). We identify these quantities with the Root Mean Square Fluctuations (RMSF) of the consecutive residues. The plots of RMSF(T) are shown in panels a (for CysZ5 and CysZ8) and b (for FC-GHK and RADA-IM) of Figure 8, respectively.

It can be seen from the figure that CysZ8 exhibits the smallest fluctuations, which is consistent with the formation of a tight amyloid-like structures. Small fluctuations from the mean positions are also observed for CysZ5, although the amyloid-like aggregate is less organized. FC-GHK exhibits moderate fluctuations amounting to 4–6 Å except for chain ends, while the biggest fluctuations are observed for RADA-IM, which seems to results from the fact that some of the chains are in the outer distorted β-barrel and some do not have organized structure (Figure 7a).

## 4. Conclusions

We tested the capability of the UNRES coarse-grained model of proteins to predict the aggregation propensity of peptides with the examples of 20 peptides of different lengths and compositions (Table 1). MREMD simulations were conducted for eight chains of each peptide in a periodic box with sides corresponding to the respective experimental concentration.

We used the experimental aggregation data of our previous work [27,31,32] of 13 peptides, while seven peptides were synthesized and investigated in this work. In all but one case (the 7–13 fragment of the Sup35 prion protein, with sequence H-GNNQQNY-NH${}_{2}$), UNRES correctly predicted the octamer dissociation temperatures to be above the experimental incubation temperature (310 K). For the three peptides that did not exhibit any association, the simulated octamer-dissociation temperatures were well below the incubation temperature (Figure 3a).

For CysZ2 and CysZ3, which reluctantly form small fibrils, the octamer-melting temperatures were close to the incubation temperature (Figure 3a). Thus, UNRES qualitatively correctly predicted the association propensity for 19 of the 20 peptides studied. For the associating peptides, except for CysZ5, UNRES predicted a considerable amount of β-structure resulting from inter-chain association.

For the 7–13 fragment of the Sup35 prion protein, UNRES predicted a low melting temperature, even though this peptide forms small fibrils and has a typical amyloidogenic sequence. One reason that UNRES does not predict its aggregation is that the sequence of this peptide contains many glutamine and asparagine residues. UNRES treats the side chains of these residues in a mean-field manner, ignoring the formation of specific hydrogen bonds between their carboxyamide groups while forming aggregates.

This result suggests that the respective sidechain–sidechain potentials in UNRES need to be revised (likely by introducing screening from the solvent and solvent polarization) to handle the aggregation of glutamine- and asparagine-rich peptides. The results obtained for the RADA-IM—RADA-KGHK peptides that begin with the (RADA)${}_{4}$ sequence, which contains eight charged residues, suggest that UNRES can also be used to predict the hydrogel-forming properties and the transition of a hydrogel into the sol phase. Below the incubation temperature, the UNRES-simulated structures of the octamers had large β-sheet sections formed between the (RADA)${}_{4}$ strands that expose the arginine and other charged side chains to the surface (Figure 7a).

Consequently, water can easily be accommodated in the form of water bridges between the layers of the β-sheets (there is a substantial amount of water-bridged arginine sidechains in proteins [65]). At higher temperatures, the β-structure becomes destroyed and the (RADA)${}_{4}$ sections form α-helices (Figure 7b), which suggests that the hydrogel melts and that micelles are formed.

## Figures and Tables

**Figure 1 biomolecules-12-01140-f001:**
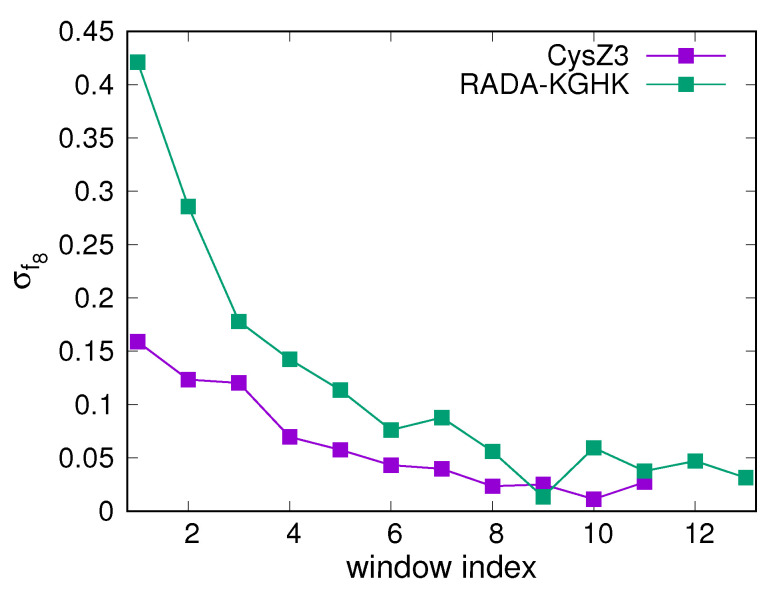
Plots of the standard deviation of the fraction of octamers ($\sigma_{f_{8}}$) of the subsequent simulation window, except from the last one, from that of the last simulation window for the CysZ3 and RADA-KGHK systems. The graph was drawn with gnuplot [57].

**Figure 2 biomolecules-12-01140-f002:**
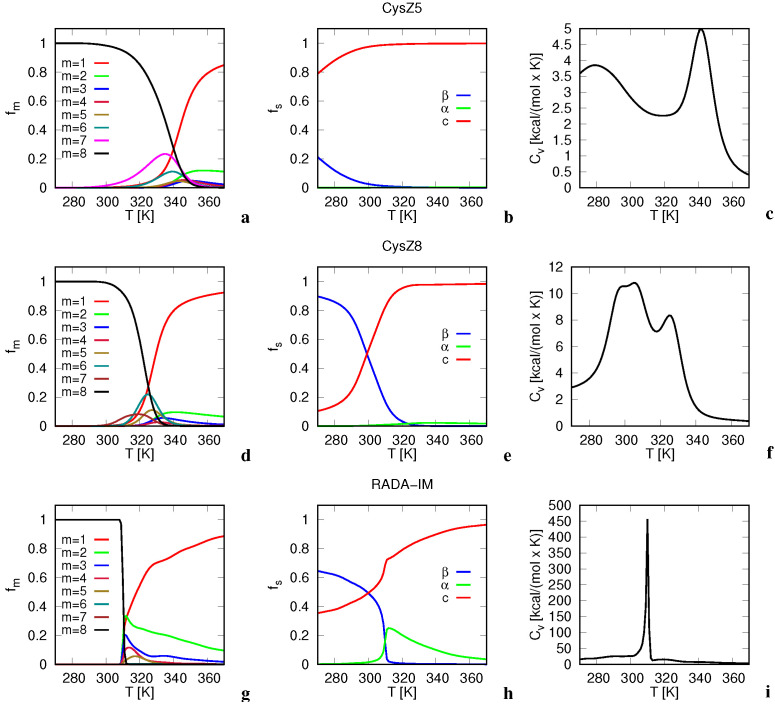
Temperature profiles of the fractions of the oligomer and subsequent multimers ($f_{m},$m=1,…,8; **a**,**d**,**g**), fractions of β, α, and statistical-coil structure ($f_{s},\;s=\alpha ,\beta ,c$; **b**,**e**,**h**) and heat capacity (**c**,**e**,**i**) of CysZ5 (**a**–**c**) CysZ8 (**d**–**f**) and RADA-IM (**g**–**i**) obtained from MREMD simulations of eight peptide chains. The peptide names are shown above the middle graphs. The graphs were drawn with gnuplot [57].

**Figure 3 biomolecules-12-01140-f003:**
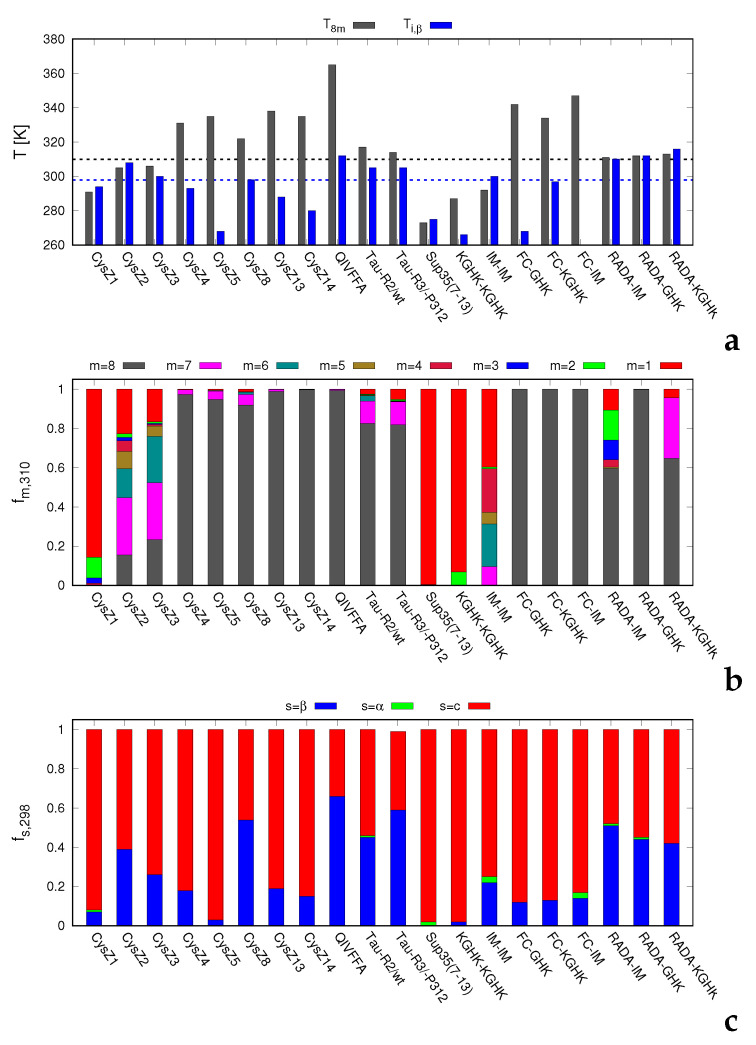
(**a**) Bar plot of the octamer half-melting temperatures ($T_{8m}$) and the temperatures at which the temperature dependence of the fraction of β-structure has an inflection point ($T_{i,\beta }$). The dashed horizontal black and blue lines correspond to the incubation (310 K) and the room (298 K) temperatures, respectively. (**b**) Stacked-histogram plot of the fractions of the monomer and the oligomers. (**c**) Stacked-histogram plot of the fractions of β, α and coil structures for all peptides studied. The graphs were drawn with gnuplot [57].

**Figure 4 biomolecules-12-01140-f004:**
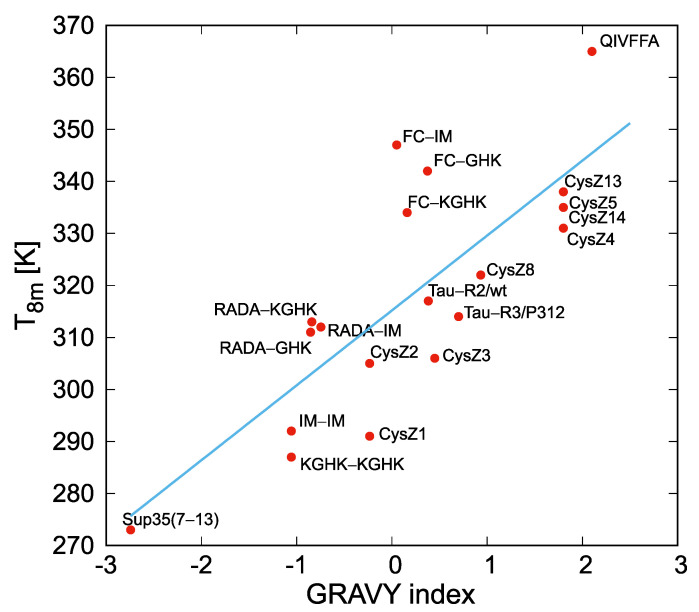
Plot of the correlation between the GRAVY indices and the half−point octamer−melting temperatures of the peptides studied. The correlation equation is $T_{8m}=14.4\times GRAVY+315$. The correlation coefficient and standard deviation in $T_{8m}$ are R=0.7946 and $\sigma_{T_{m}}=13.5$ K, respectively. The graph was drawn with gnuplot [57].

**Figure 5 biomolecules-12-01140-f005:**
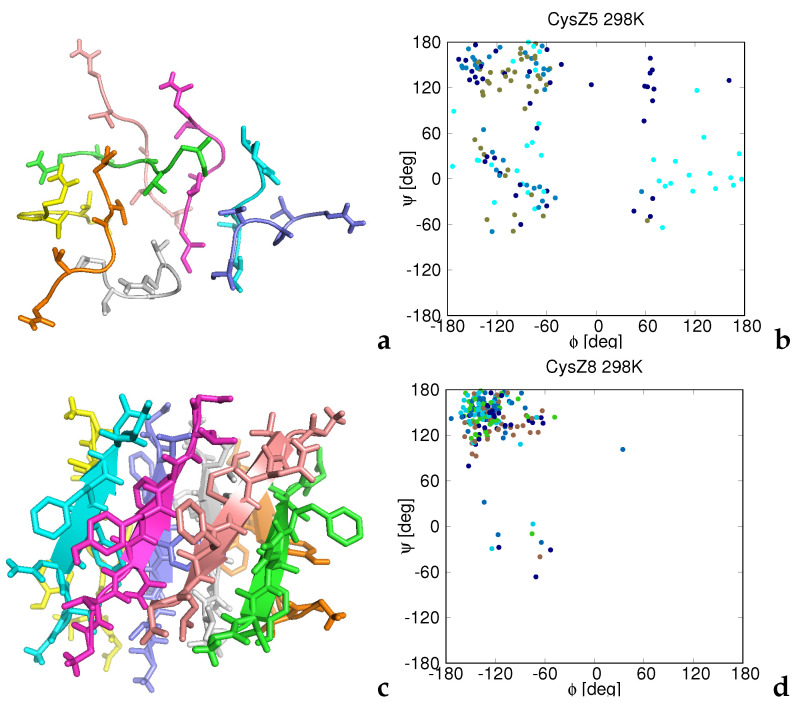
Representative structures of the octamers of CysZ5 (**a**) and CysZ8 (**c**) at *T* = 298 K obtained in MREMD simulations of eight-chain systems and the corresponding plots of the backbone ϕ and ψ angles collected from all five models obtained by the cluster analysis at this temperature (**b**,**d**). Each chain in panels **a** and **c** is colored by a different color. Backbones are represented as ribbons and the side chains are shown in the stick representation. The structures were obtained by converting the UNRES chains into the all-atom representation. Each point in the (ϕ,ψ)-plots is colored according to residue index (chain index ignored), from blue to red from the N- to the C-terminus. The drawings were produced with PyMol [64] and gnuplot [57], respectively.

**Figure 6 biomolecules-12-01140-f006:**
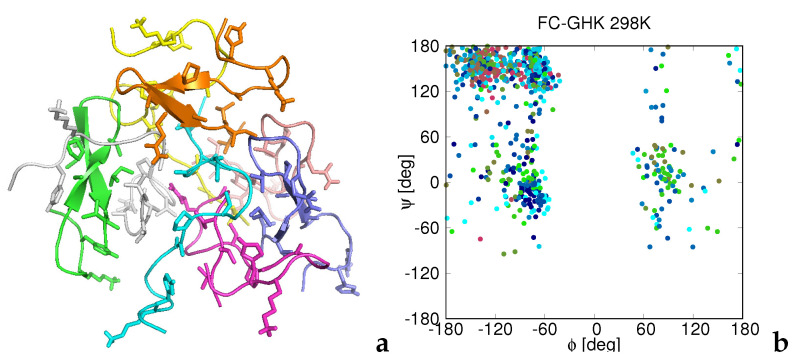
A representative structure of the octamer of FC-GHK at *T* = 298 K obtained in MREMD simulations of the corresponding eight-chain system (**a**) and a plot of the backbone ϕ and ψ angles collected from all five models obtained by the cluster analysis at this temperature (**b**). Each chain in panel a is colored by a different color. Backbones are represented as ribbons and the side chains are shown in the stick representation. The structures were obtained by converting the UNRES chains into the all-atom representation. Each point in the (ϕ,ψ)-plots is colored according to residue index (chain index ignored), from blue to red from the N- to the C-terminus. The drawings were produced with PyMol [64] and gnuplot [57], respectively.

**Figure 7 biomolecules-12-01140-f007:**
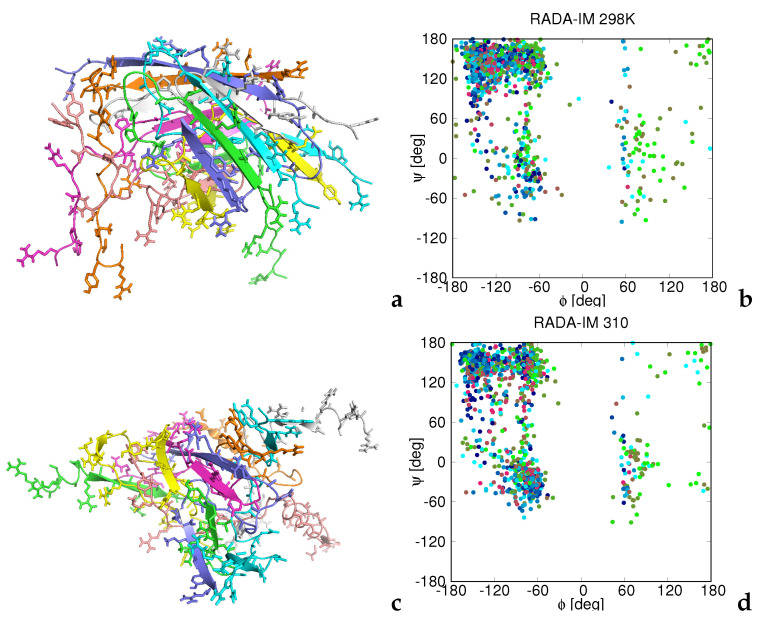
Representative structures of the octamers of RADA-IM at *T* = 298 K (**a**) and *T* = 310 K (**b**) obtained in MREMD simulations of eight-chain systems and the corresponding plots of the backbone ϕ and ψ angles collected from all five models obtained by the cluster analysis at these two temperatures (**b**,**d**). Each chain in panels a and c is colored by a different color. Backbones are represented as ribbons and the side chains are shown in the stick representation. The structures were obtained by converting the UNRES chains into the all-atom representation. Each point in the (ϕ,ψ)-plots is colored according to residue index (chain index ignored), from blue to red from the N- to the C-terminus. The drawings were produced with PyMol [64] and gnuplot [57], respectively.

**Figure 8 biomolecules-12-01140-f008:**
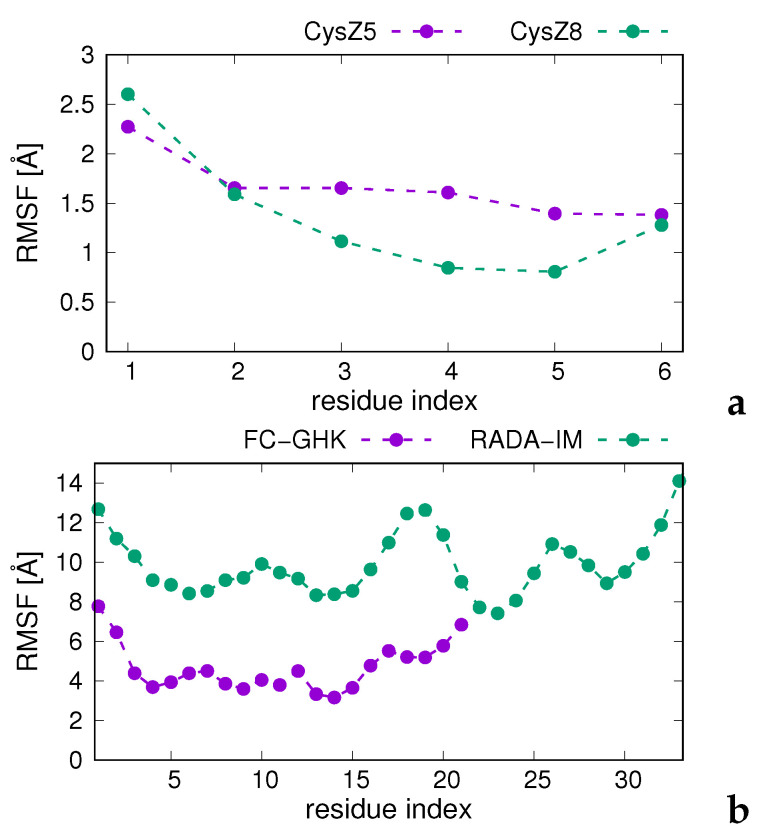
C${}^{\alpha }$-atom root mean square fluctuations (RMSF) for CysZ5, CysZ8, FC-GHK and RADA-IM plotted in residue index. The graphs for CysZ5 and CysZ8 are shown in panel (**a**), while those for FC-GHK and RADA-IM are shown in panel (**b**). The RMSFs were calculated as the standard deviations of the positions of the C${}^{\alpha }$ atoms over all five models obtained in MREMD simulations of eight-chain systems and subsequent cluster analysis at *T* = 298 K and all chains from those of the mean chain structure The graphs were drawn with gnuplot [57].

**Table 1 biomolecules-12-01140-t001:** The peptides studied in this work.

Short Name ${}^{a}$	n ${}^{b}$	Sequence	Q ${}^{c}$	GRAVY ${}^{d}$	C [mM] ^e^	l [Å] ${}^{f}$	Assoc. ${}^{g}$	Source
CysZ1	6	H-ARKQIV-NH${}_{2}$	3.0	−0.233	4.210	146.7	no	[27]
CysZ2	6	H-RKQIVA-NH${}_{2}$	3.0	−0.233	4.210	146.7	sf	[27]
CysZ3	6	H-KQIVAG-NH${}_{2}$	2.0	0.450	4.888	139.5	sf	[27]
CysZ4	6	H-QIVAGV-NH${}_{2}$	1.0	1.800	2.565	173.0	f	[27]
CysZ5	6	H-IVAGVN-OH	1.0	1.800	5.257	136.2	sf	[27]
CysZ8	6	H-GVNYFL-NH${}_{2}$	1.0	0.933	2.110	184.6	f	[27]
CysZ13	6	H-QAGIVV-NH${}_{2}$	1.0	1.800	1.710	198.0	f	[27]
CysZ14	6	H-QIGVAV-NH${}_{2}$	1.0	1.800	1.710	198.0	f	${}^{h}$
QIVFFA	6	H-QIVFFA-NH${}_{2}$	1.0	2.100	1.384	212.5	f	${}^{h}$
Tau R2/wt	6	H-VQIINK-NH${}_{2}$	2.0	0.383	5.784	131.9	f	[28] ${}^{h}$
Tau R3/-P312	6	H-VQIVYK-NH${}_{2}$	2.0	0.700	1.337	215.0	f	[28,45] ${}^{h}$
Sup35(7-13 )	7	H-GNNQQNY-NH${}_{2}$	1.0	−2.743	1.196	223.1	sf	[29,30] ${}^{h}$
KGHK-KGHK	18	Ac-KGHKGGGAAPVGGGKGHK-NH${}_{2}$	4.2	−1.056	0.609	279.3	no	${}^{h}$
IM-IM	22	Ac-RDKVYRGGGAAPVGGGRDKVYR-NH${}_{2}$	4.0	−1.055	0.421	316.0	no	${}^{h}$
FC-GHK	19	Ac-GHKGGGAAPVGGGQAGIVV-NH${}_{2}$	4.0	0.374	0.614	278.7	f	[31]
FC-KGHK	20	Ac-KGHKGGGAAPVGGGQAGIVV-NH${}_{2}$	2.1	0.160	0.569	285.8	f	[31]
FC-IM	22	Ac-RDKVYRGGGAAPVGGGQAGIVV-NH${}_{2}$	2.0	0.050	0.470	304.5	f	[31]
RADA-IM	32	Ac-(RADA)${}_{4}$-GGGAAPVGGGRDKVYR-NH${}_{2}$	2.0	−0.853	3.116	162.1	Fg	[32]
RADA-GHK	28	Ac-(RADA)${}_{4}$-GGGAAPVGGGHK-NH${}_{2}$	1.1	−0.746	3.763	152.3	Fg	[32]
RADA-KGHK	30	Ac-(RADA)${}_{4}$-GGGAAPVGGGKGHK-NH${}_{2}$	2.1	−0.840	3.518	155.7	Fg	[32]

*^a^* CysZ1–CysZ5, CysZ8, CysZ13 and CysZ14: peptides excised from the human cystatin *β*-sheet or loop regions; Tau R2/wt and Tau R3/-P312: peptides excised from the Tau protein; Sup35(7-13): the 7–13 fragment of the prion Sup35 protein; KGHK-KGHK and IM-IM: signaling peptides; FC-GHK, FC-KGHK, FC-IM: functionalized human cystatin fragments that form wound-healing films; RADA-IM—RADA-KGHK: potentially aggregating sequences merged with charged sequences, which are signaling peptides that form gels; *^b^* Number of residues; *^c^* Total charge at pH = 7, estimated using the peptcalc.com utility [42]; *^d^* Grand hydropathy index, estimated based on the Kyte and Doolitle hydropathic indices [43], using the ExPasy [44] protparm tool (https://web.expasy.org/protparam/ accessed on 8 August 2022) ; *^e^* Experimental concentration; *^f^* Cubic box-side length applied in simulations, corresponding to the experimental peptide concentration; *^g^* Form of association; no: no association; f: fibrils; sf: small fibrils; Fg: the peptides form gels (immediately after dissolution), which show fiber structure in TEM images; ^*h*^ This work.

**Table 2 biomolecules-12-01140-t002:** Fractions of secondary structure of the solutions of the peptides studied before and after incubation determined from CD measurements.

Peptide	Before Incubation					After Incubation			
	$f_{e}$ ${}^{a}$	$f_{\alpha }$ ${}^{a}$	$f_{t}$ ${}^{a}$	$f_{c}$ ${}^{a}$		$f_{e}$ ${}^{a}$	$f_{\alpha }$ ${}^{a}$	$f_{t}$ ${}^{a}$	$f_{c}$ ${}^{a}$
CysZ1	0.36	0.05	0.19	0.40		0.37	0.05	0.19	0.39
CysZ2	0.37	0.05	0.19	0.39		0.37	0.05	0.19	0.39
CysZ3	0.38	0.05	0.20	0.37		0.39	0.05	0.19	0.37
CysZ4	0.37	0.05	0.20	0.38		0.59	0.07	0.20	0.14
CysZ5	0.48	0.03	0.19	0.30		0.44	0.03	0.21	0.32
CysZ8	0.40	0.04	0.21	0.35		0.47	0.04	0.21	0.28
CysZ13	0.37	0.05	0.20	0.38		0.44	0.04	0.20	0.33
CysZ14	0.33	0.21	0.17	0.29		0.30	0.19	0.20	0.29
QIVFFA	0.59	0.01	0.18	0.22		0.59	0.01	0.18	0.22
Tau R2/wt	0.53	0.02	0.19	0.26		0.53	0.02	0.19	0.26
Tau R3/-P312	0.41	0.04	0.21	0.34		0.46	0.03	0.20	0.31
Sup35(7-13)	0.42	0.05	0.20	0.33		0.42	0.04	0.21	0.33
KGHK-KGHK ${}^{b}$	0.25	0.06	0.23	0.46		–	–	–	–
IM-IM ${}^{b}$	0.47	0.02	0.21	0.30		–	–	–	–
FC-GHK	0.38	0.04	0.20	0.38		0.34	0.10	0.26	0.30
FC-KGHK	0.44	0.04	0.20	0.32		0.39	0.05	0.21	0.35
FC-IM	0.40	0.05	0.20	0.35		0.49	0.03	0.20	0.28
RADA-IM ${}^{c}$	0.48	0.05	0.21	0.26		–	–	–	–
RADA-GHK ${}^{c}$	0.45	0.06	0.22	0.27		–	–	–	–
RADA-KGHK ${}^{c}$	0.42	0.06	0.22	0.30		–	–	–	–

*^a^* Fractions of extended (e), *α*, turn (t) and statistical-coil (c) structure, respectively, estimated based on CD spectra using the CONTIN program of the CDPro package [46]; *^b^* CD measurements were not performed after incubation; *^c^* These peptides form gels almost immediately, and therefore the CD measurements were made for gels.

## Data Availability

The simulated structures of the octamers are available from the authors.

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
