# Peer review of "Prediction of Aggregation of Biologically-Active Peptides with the UNRES Coarse-Grained Model"

_biomolecules, 2022, doi:10.3390/biom12081140_

Round 1

Reviewer 1 Report

In the present manuscript, Biskupek et al. investigated the association of 20 peptides with size from 6 to 32 amino-acid residues using UNRES coarse-grained model. They predicted the association propensity of 19 out of 20 peptides studied qualitatively correctly. For the associating peptides, UNRES also predicts a considerable amount of β-structure. In general, the paper is well organized, and the conclusions are convincing. The reviewer would like to recommend the paper for publication after the minor revision.

1. Some indicators are needed to demonstrate that the MREMD simulations have converged.

2. There are eight peptides in the simulation box and also a corresponding experimental concentration for each system. How did the authors achieve consistent concentrations in the simulation and experiment, by changing the size of the box even with implicit water?

3. How did the authors judge the size of the oligomers, that is, what is the criterion for the association between chains? The hydrogen bonding network of main chains is suggested to be taken into consideration in the criteria, as this is an important basis for distinguishing the fibril-like or β-sheet-rich aggregates. These need to be further clarified in the manuscript.

4. Temperature profiles of fm and fractions of secondary structures for RADA-IM are interesting and seem to imply a first-order phase transition. The reviewer suggested the authors to calculate the heat capacity or other order parameters to verify it.

5. The results involving amino acids N and Q deviate from the experimental data, suggesting that the polarizing effect of the solvent may be underestimated. The authors would further discuss how to revise the UNRES model at this point.

6. Some recent references regarding the aggregation of amyloid peptides would be cited to further explain the key interactions that drive protein aggregation. These relevant references include Quant. Biol., 2022, 10(1):17; Phys. Chem. Chem. Phys., 2021, 23:20615; Adv. Mater. 2020, 32:1901690.

Round 2

Reviewer 2 Report

The authors have satisfactory addressed the comments that I had. I feel based on my level of expertise that the manuscript in its current form can be published without changes.

Author Response

Thanks.